# Comparative Assessment of the Influence of Various Time Intervals upon the Linear Accuracy of Regular, Scannable, and Transparent Vinyl Polysiloxane-Based Bite Registration Materials for Indirect Dental Restoration Fabrication

**DOI:** 10.3390/polym17010052

**Published:** 2024-12-28

**Authors:** Firas K. Alqarawi, Bandar M. A. AL-Makramani, Praveen Gangadharappa, Khurshid Mattoo, Maryam Hadi, Mohammad Alamri, Ebrahim Fihaid Alsubaiy, Saeed M. Alqahtani, Mohammed E. Sayed

**Affiliations:** 1Department of Substitutive Dental Sciences, College of Dentistry, Imam Abdulrahman Bin Faisal University, P.O. Box 1982, Dammam 31441, Saudi Arabia; fkalqarawi@iau.edu.sa; 2Department of Prosthetic Dental Sciences, College of Dentistry, Jazan University, Jazan 45142, Saudi Arabia; makramani@yahoo.com (B.M.A.A.-M.); praveengmds@gmail.com (P.G.); drkamattoo@rediffmail.com (K.M.); 3Primary Care Administration, Ministery of Health, Jazan 45911, Saudi Arabia; maryam_hadi@outlook.sa; 4Department of Restorative Dental Sciences, College of Dentistry, King Khalid University, Abha 62529, Saudi Arabia; moabalamri@kku.edu.sa; 5Department of Prosthetic Dental Sciences, College of Dentistry, King Khalid University, Abha 62529, Saudi Arabia; alsbuay@kku.edu.sa (E.F.A.); smaalqahtani@kku.edu.sa (S.M.A.)

**Keywords:** jaw relation records, polymeric bite registration materials, polyvinyl siloxane, therapeutic occlusion, bite record storage

## Abstract

Interocclusal records (IORs) created with bite registration materials (BRMs) accurately reflect the opposing teeth’s physiological and anatomical associations in digital and traditional dentistry. This study assessed the linear dimensional accuracy of vinyl polysiloxane-based scannable and transparent BRMs over obligatory clinical time intervals (1, 24, 72, and 168 h/s). A total of 3 scannable [Flexitime Bite, Occlufast CAD, Virtual CADBite] and 3 transparent [Maxill Bite, Charmflex Bite, Defend ClearBite] VPS-based BRMs were divided into 28 subgroups by time interval: 1, 24, 72, and 168 h/s. Stereomicroscope measurements of 420 standardised disk-shaped specimens with three distinct linear distances between crossing vertical and horizontal lines were taken. Comparisons with the conventional BRM determined the scannable and transparent BRMs’ accuracy, while comparisons with die dimensions yielded linear dimensional changes. Statistical analysis used median rank scores, interquartile range, and median. Using a one-way ANOVA rank and Dunn test, differences were assessed between and within groups at a probability ‘*p*’ value of 0.05 (*p* ≤ 0.05). Mean linear dimensions for CAD and transparent IOR materials were [−0.06 (−0.24%) to −0.15 (−0.6%)] and [−0.06 (0.24%) to −0.10 (0.40%)] millimetres, respectively. Virtual CADBite and Maxill Bite had the lowest linear disagreement after 1 h, but both showed significant variations at 7 days. Other commercial brands maintained their clinically acceptable linear accuracy (0.11). Flexitime Bite (CAD) was the sole material with a linear accuracy above the clinical threshold. IOR shrinkage reduced the linear dimensions in all materials. Until 7 days, all IOR materials except Flexitime bite (CAD) were clinically correct. Virtual CADBite and Maxill bite changed significantly during 1 h and 7 days.

## 1. Introduction

An interocclusal record (IOR) (synonym, bite registration record (BRR)) defines both the static and dynamic relationships of teeth in relation to adjacent and opposing teeth arches, jaws, and to the cranium. Digital methods of jaw relation and bite registration records (virtual and physical) have not only paved the way for conventional materials but have also diminished human-induced errors in such critical dental restorative procedures [1]. Errors in occlusion have also become easier to locate with the use of various scanning devices (T scan, intraoral scans) [2]. Among various currently available digital jaw relation procedures, one technology mounts a CADCAM generated dental cast (virtual cast) using algorithms (best fit alignment), thereby eliminating the need for a physical IOR [3]. Intraoral scanning and scanning of patients’ casts/models and/or IORs can also produce virtual casts [4]. When CADCAM made physical casts are to be mounted, scanning of the buccal surfaces of the maxillary/mandibular teeth in maximum intercuspation provides data that are analysed with software [3]. Occlusal contacts provided from virtual occlusion are accurate in addition to providing objective data like occlusal timings, contact sequences, and quantity of force [5]. Virtual IOR (iTero Element scanner, Align Technology, Inc., Tempe, AZ, USA) is another technology that has shown promising results in terms of accuracy and reproducibility when compared with polyvinylsiloxane (PVS) physical records [6]. Currently, however, the most common digital technique utilises scanning of a polyvinylsiloxane IOR in the intercuspal position, which provides a two-dimensional image that is analysed with image computer software [7,8]. Multiple studies have reported this technique to be highly reliable and valid for determining occlusal contacts [9,10,11], so it has thus been considered a standard method for digital jaw relations. Irrespective of the technique, the digital static occlusal analysis is accomplished in three steps: patient closing in maximum intercuspation on an indicator (sensor, silicone material, or articulation indicator), interpretation of the IOR on a computer, and finally, storage and transfer of the IOR. All different indicators have been found to have high reliability and validity [12,13]. The digital articulator system that is required for mounting these records, however, have been found to be less accurate than the conventional articulator system [14], with VPS being more accurate than wax in either the conventional or digital articulator system. In another study, digital scans (t scan (Tekscan)/3D intraoral scan) were found to be less reliable in measuring occlusal contact area when compared with occlusal registration [9]. The size of the occlusal contact and the intensity of the contact are also significant factors in determining the amount of correction required in addition to the accuracy of the working and non-working casts [15]. IOR, irrespective of the material used, can provide information about the contact intensity, which has been reported to be less precise when virtual occlusal records are used [16]. Despite less variance with scanning IORs than conventional physical IORs, there are certain limitations with the use of scanning, especially the use of intraoral scanners. They have trouble creating accurate virtual images and scanning entire dental arches or edentulous arches [17,18]. To reduce background noise and scanning errors, titanium dioxide powder should be applied to objects, but different application distances and times can cause coating thickness variations, causing scan errors [19]. Liquid-type scanning-aid agents and powder-free intraoral scanners have advantages, but the optical properties can be affected by environmental factors [4,11], and powder coatings can improve accuracy [20]. Longer scans, like full-arch scans, may result in enhanced errors due to matching or stitching errors [4,14], which increase with scan length [11]. Intraoral scanners use single images stitched together to create 3D models [21].

Polyether- and VPS-based bite registration materials (BRMs) are increasingly favoured for their handling properties, precision, and durability [8,18]. They resemble dental impression materials but exhibit changed qualities due to less well-known adjustments in the mix of plasticisers and catalysts [22]. These materials, however, necessitate a carrier and are economically viable. The set material possesses sufficient elasticity for removal from the mouth without deformation [6], while simultaneously exhibiting enough rigidity to enable it to withstand any deformation under applied stresses [7,12]. They are basically a composite of different elemental compositions in which Si, C, and O elements are higher than Al, Ti, Ca, Na, Mg [23]. VPS BRMs are essentially particle-reinforced composites within the organic (vinyl polysiloxane) matrix, rendering them multiphase (two or more) phase systems, with the difference in phases due to the mean atomic number distribution and contrast [12,23]. SiO_2_ or silicate-glass filler particles, as well as the vinyl-polysiloxane matrix, are responsible for the high C, Si, and O content observed in all products [14]. All scannable IBRR materials have a higher TiO_2_ concentration to add to the material’s high reflectivity [13], which is intended to meet the criteria for CADCAM optical scanning [13,23]. As a non-reinforcing filler, CaCO_3_ is utilised to enhance dispersion, processing, and extrudability and to regulate the viscosity and sagging of the finished product, together with Al, Mg, and Zn oxides or various salts [18,23]. Setting of the matrix also results in the production of hydrogen, which is absorbed by the alkaline CaCO_3_ [24,25]. In evaluating a material’s scannability, contrast and brightness are the two most critical factors, each ideally set at 100% [26]. Intraoral recording materials are enhanced with pigments that improve contrast and brightness levels, thereby increasing their scannability [14,19]. This enhances the dynamic value of the material’s scannability [26]. Metal oxides, hydroxides, metal oxide hydrates, metal carbonates, or metal sulphates are incorporated into interocclusal recording materials at a concentration ranging from 15 to 80 weight percent to improve the brightness of these materials [23]. To enhance the material’s contrast, incorporate black or black-greyish pigments at concentrations between 0.01 and 0.0001 weight percent [26]. Suitable additives include metals, carbon, metal oxides, metal hydroxides, metal oxide hydrates, metal silicates, sulphur-containing metal silicates, metal sulphides, metal cyanides, metal selenides, metal chromates, or organic dyes [24,26,27]. Many clinical [27,28] and in vitro research studies [29,30,31,32,33,34,35,36] have revealed that the PVS-based BRMs have higher dimensional stability than other BRMs. Furthermore, noted in these investigations are PVS BRMs’ greater dimensional stability than polyether-based BRMs [29,30,31,32,33,34,35,36]. Polyether, however, was reported by Michalakis KX et al. [37], Tejo SK et al. [38], Pokale SV et al. [39], and Sonkesriya S et al. [40] to be more dimensionally accurate initially, with few of these studies indicating that PVS continues to maintain its accuracy better than polyether at subsequent time intervals (48 and 72 h) [37,39]. Sharma A et al. [41] and Lozano F et al. [42] both noted, at the same time, PVS BRMs to be more dimensionally stable than polyether. Lozano F. et al. [42] evaluated the dimensional correctness of three IORs (Aluwax, Godiva [thermoplastic bar], Occlufast Rock [VPS], and Futar D (injectable silicones), finding that although Occlufast silicone remained stable for as long as 7 days, Futar D was clinically viable for 22 days. In addition to these studies evaluating the linear accuracy, investigators have also evaluated the vertical accuracy of different BRMs in clinical [28] and in vitro [26,30,34] settings. Polyvinylsiloxane BRMs were found to be the most stable in both the vertical and lateral axis. Conventional (Registrado Xtra, Futar D Fast, and O-Bite) and scannable (Registrado Scan, Futar Cut & Trim Fast, and O-Bite Scan) BRM materials were investigated recently by Yazigi C et al. [26] for their ability to record maxillary–mandibular relationships (vertical accuracy) and their dimensional stability after one hour and forty-eight hours of storage. Scannable materials had far fewer discrepancies than conventional ones, according to the results, which demonstrated a notable difference in vertical discrepancies between the two materials. After 1 hour, the median vertical discrepancy varied between −2 μm (FS) and 11 μm (O-Bite), reaching 13 μm (Registrado Xtra and O-Bite) after 48 h, and 3 μm (Futar Cut & Trim Fast) after 48 h. After 48 h, there was an increase in the number of materials that showed clinically acceptable disparities. The accuracy of conventional and digital systems in locating occlusal contacts was examined in a recent study by Rovira-lastra B et al. [43], who discovered that Occlufast Rock (regular BRM) achieved an agreement in occlusal contact location between sessions of 85–95%. In comparison, Occlufast CAD, 200 μm articulating film, and T-Scan offered agreements of 79–86%, 68–75%, and 65–75%, respectively. In addition to scannable BRMs, transparent BRMs are also available that overcome the drawbacks of conventional opaque BRMs in that voids and bubbles introduced within the BRR are visible; these are a major source of clinical repetition and inducing iatrogenic errors in patients’ cast mounting. Compared with scannable BRMs, they contain a vinyl-terminated polydimethylsiloxane and replace silicon dioxide with quartz silica as major components. A previous study reported high tensile strength and high elastic moduli, which are critical requirements for BRMs [44]. Errors in mounting can be either vertical or horizontal (lateral or anteroposterior). While both are critical, horizontal errors are more sensitive, as a minor error can result in misfit of the casts on the BRR. The influence of vertical errors, at the same time, can be minimised by facebow use. The permissible errors in mountings have also been studied, and the threshold in such transfers has been estimated to be in the range of 0.07 mm to 0.11 mm (anteroposterior and lateral) and less than 1 mm vertically [45,46].

This study, therefore, aimed to evaluate the influence of various time intervals upon the linear accuracy of regular, scannable, and transparent vinyl polysiloxane-based bite registration materials for indirect dental restoration fabrication. The main objective of the study was to find whether scannable and transparent BRMs are as accurate as regular BRMs through clinical processing time (1 h) and various recommended laboratory processing times (24, 72, 168 h). Also, the study results were aimed at finding whether these changes are acceptable within the clinical threshold of predefined limits in terms of linear changes within a BRR.

## 2. Materials and Methods

### 2.1. Ethics

Following the guidelines laid out by the organisation, written ethical permission was acquired for the conduct of this study under registration number (Ref. No. CODJU-2326I). All substances under investigation have been duly approved by both international and local drug organisations, and they are all biocompatible with humans.

### 2.2. Study Design

Control and test specimens were randomly assigned to different groups in this in vitro investigation, which used a comparative multiple-group experimental design approach. The time intervals (1 h; 1, 3, and 7 days) [38,39,41] and the BRR materials (CAD and transparent) were considered independent variables, whereas the linear accuracy (in millimetres and percentage) was the dependent variable. Several steps were included in the experiment, including machining a stainless-steel standard three-component die, assembling the mould, preparing the specimen, and measuring surface markings that indicated linear accuracy.

### 2.3. Operational Definitions [47]

Records of the static or dynamic relationship between two teeth, arches, or jaws that are opposing are referred to as a BRR (synonym bit, occlusal, or interocclusal record) [47]. By transferring maxillomandibular relationships from the mouth to the articulator, the BRR ensures horizontal stability and prevents rotation or translation of the cast by capturing the position of opposing teeth. BRR types include centric and eccentric, which further are divided into protrusive and right and left lateral BRRs.

### 2.4. Materials

Three of the six BRMs studied belonged to the scannable (CAD) category, while the other three were transparent; a brief summary of each is provided in Table 1. The basic ingredient of all BRMs is vinyl polysiloxane elastomers; the only changes are in the composition and percentages of the individual components. Each brand’s manufacturer-recommended clinical criteria for the amount of time spent mixing, manipulating, working, and setting were adhered to [48,49,50,51,52,53] (Table 1).

### 2.5. Sample Size

The research was structured with two primary experimental groups (material type based), each of which used three distinct commercially available BRMs. There were four subgroups for each commercial BRM according to the time interval: control (1 h) and 1, 3, and 7 days. With a power assumption of 80%, a type 1 error rate of 0.05, and an effect size of 0.28, the predicted total sample size was 420 specimens, with a minimum of 15 specimens in each subgroup. The results were collected from the Nquery program (v7.0; Informer Technologies, California, Los Angeles, CA, USA), which determined the sample size using the formula [(N = 2 σ2 × (Z α + Z β) 2/2)] [54].

### 2.6. Standardization of Specimen and Preparation (Figure 1)

Multiple clinical guidelines agree that three millimetres is the optimal thickness for a BRR. In that context, a three-piece (ruled cylinder, mould spacer, riser) stainless-steel (austenitic) die was fabricated according to standard specifications for measuring dimensional accuracy of dental elastomers used to make impressions (American Dental Association specification no. 19) [55]. Figure 1 shows the components, engraved vertical and horizontal lines, coordinates, and dimensions of the multipiece standardised die. When assembled together, the mould forms a 3 mm uniform space for BRM with engraved lines transferred to the material specimen. The linear accuracy of test specimen was measured at three different intersections or coordinates (p1–p2, p3–p4, p5–p6) between vertical (cc′, dd′) and horizontal lines (X, Y, and Z). Distance between three coordinates was uniformly placed at 25 mm, the average of which served as control. A thickness of 3 mm was ensured by the mould spacer, which was 6 mm in height, 3 mm of which were embedded into the shoulder of the cylinder block and 3 mm left for material space. A riser in the form of a 33 mm stainless steel sheet was fabricated for the purpose of removal of the specimen from the mould. Distance between three horizontal lines was standardised at 2.5 mm.

**Figure 1 polymers-17-00052-f001:**
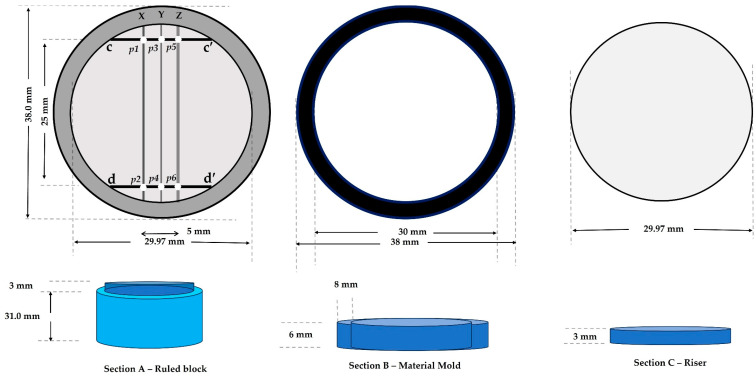
Components and dimensions of a three-unit standard stainless-steel die and the coordinates used for measuring linear accuracy.

### 2.7. BRM Dispensing

Each BRM that was studied is currently on the commercial market in the form of a standard 100 mL two-conjoined-cartridge system, with one cartridge serving as a base and the other as a catalyst, all placed on a sterile dispenser [48,49,50,51,52,53]. The dispenser is hydraulicly shaped and operates similarly to a toy gun. An autoclave cycle (pre-vacuum, 3 min contact, 132-degree centigrade heat, 20 min cool time) for complete sterilization and a chemical mixture (15% isopropyl alcohol, 0.3% ammonium salt) for disinfection are standard manufacturers’ recommendations before cartridge exchange and each use, respectively. Each brand was stored, manipulated, and changed according to the manufacturer’s instructions, which typically included insignificant temperature differences. An automix plastic tip is secured to the cartridge system using a key–keyway lock mechanism, which joins the individual cartridges that contain catalyst and base VPS pastes at the front. Two separate plungers, one for each cartridge, are housed in the dispenser. Squeezing the trigger causes the plungers to go forward, which forces the material through the automix tip and allows the gun to dispense equal amounts. The automix tip uses its length to combine the two components, and then it may be used to apply a clean mixture directly to the affected area [48,49,50,51,52,53].

### 2.8. Specimen Preparation/Grouping/Subgrouping (Figure 2)

Specimens were prepared for each BRM subgroup by placing the constructed mould on top of the dispenser and releasing material. After filling, any surplus was removed with a polythene laminated glass slab that also acted as a model for the occlusion compression pressure (0.5 kg gram load) [33,37]. All materials were subjected to the load at the times specified in Table 1 (setting times). In order to replicate the conditions found in the mouth, the mould containing the load was immersed in a water bath that was thermostatically controlled (37 ± 1 degrees Celsius) until the materials had set. After removing each specimen from the mould, any extra was either thinned out or polished off. Thus, all samples were uniformly 30 mm in diameter and 3 mm thick, with the following inscribed lines: X, Y, Z, cd, c′d′, and coordinates (p1–p2, p3–p4, p5–p6). After soaking each piece of tissue in a 0.5% glutaraldehyde solution for 10 min, we followed the same protocol as the clinical recommendation [56]. After that, according to the specified groups and time intervals, washing, drying, and storing were carried out. The specimens for each subgroup were kept in a sealed, humidity-free polythene bag and kept at room temperature until their measurements were taken. Thus, 60 specimens were prepared for regular (Occlufast rock) with four subgroups (1 h, 24 h, 72 h, 168 h), each subgroup serving as control for test materials (scannable, transparent). For experimental subgroups, 180 specimens each were prepared for 3 scannable (Flexitime CAD Bite (SF1, SF24, SF72, SF168), Occlufast CAD (SO1, SO24, SO72, SO168), and Virtual CADBite (SV1, SV24, SV72, SV168)) and transparent (Charmflex Bite (TC1, TC24, TC72, TC168), Defend Clearbite (TD1, TD24, TD72, TD168), and Maxill Bite (TM1, TM24, TM72, TM168)) BRMs, with 4 time-interval subgroups (1 h, 24 h, 72 h, 168 h). A total of 28 subgroups, with each subgroup having 15 specimens, were measured for linear accuracy.

**Figure 2 polymers-17-00052-f002:**
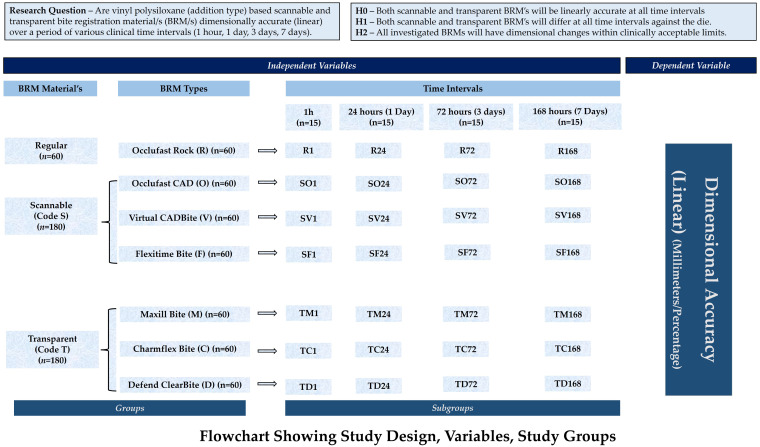
Study flow chart showing materials, variables, groups, and subgroups.

### 2.9. Measurements

From one side’s point of intersection (cc′) to the other side (dd′) in three coordinates (p1–p2, p3–p4, p5–p6), a stereomicroscope was utilised to take three linear measurements, yielding three readings per specimen. A single operator, previously calibrated to the usage and measurement of the stereoscope, measured all observations. Using the stage plate to orient the specimens’ flat surfaces, we then illuminated them using upper and lower lights and controlled the camera, which was connected via a USB CCD. After adjusting and focussing the eyepiece dioptre on a regular basis, all specimens were inspected with conventional settings (10× magnification) until the marking lines (edges of the line) were clearly apparent. The mean of each specimen was obtained by averaging three measurements between these coordinates: p1–p2, p3–p4, p5–p6.

### 2.10. Statistical Analysis

Before running the statistical analysis in Statistical Package for the Social Sciences (SPSS, Version 25, IBM Corp., Armonk, NY, USA), all of the raw data from each subgroup were imported into MS Excel (version 20H2) to be cleaned, standardised, and coded. The main source of statistical interpretation between the experimental and control groups was in the form or median values, interquartile range, and mean rank scores (MRSs). the dimensional changes were expressed in terms of distance in millimetres and frequency distribution (percentage) when compared with the original distances on the die. Normality tests (Shapiro–Wilk) determined the distribution of data, which indicated the use of median values and mean rank scores to avoid errors induced by asymmetric data distribution. A one-way ANOVA rank test (Kruskal–Wallis) was used to analyse the differences between control (regular) and experimental BRMs subgroups (scannable and transparent) using their mean rank scores, which provides a more sensitive differential. The differences within subgroups at individual time intervals were performed using post hoc multiple comparison Dunn test after correcting the probability value. The correction was performed by using the formula (Corrected α = α/m), where α denotes the set probability value (0.05) and m stands for the number of total subgroups. A primary arithmetic operation in the form of subtraction (X − Y) gave linear dimensional changes between each subgroup value (in millimetres) (Y) and the original dimension on the die (Y), thus describing the average physical change in each material subgroup at different time intervals. Increase in dimensions was considered to be positive (+), while decreased dimensions indicated negative (−). The same numerical equations also derived the total dimensional change expressed in percent using the following mathematical formula: D (dimensional change).% = [(X − Y)/X × 100], where X and Y values are the same as in the previous section [38,39].

## 3. Results

### 3.1. Regular (Occlufast Rock) BRM

Table 2 presents the comparative differences in the median values and their interquartile ranges for the control and experimental VPS BRM (scannable and transparent) subgroups based on time intervals. Since all subgroups of this material served as the control to determine whether scannable and transparent BRMs are equally accurate, results show that all commercial brands of both scannable and transparent BRMs are more accurate than the regular BRM. Gp SF1 was the only material that had linear changes that were similar to the regular at 1 h, it but showed lesser changes at subsequent time intervals of 24, 72, 168 h.

### 3.2. Scannable BRM (Occlufast CAD, Virtual CADBite, Flexitime Bite)

All three scannable BRMs showed fewer linear changes than the regular BRM at all time intervals, with Virtual CADBite not only showing the fewest changes at all time intervals but showing no change between 1 and 24 h. Flexitime bite (Mdn (IQR), 24.89 (0.09) at 1 h to 24.86 (0.08) at 172 h) showed more changes among the scannable BRMs at all time intervals, but apart from the 1 h, it showed fewer changes than the regular BRM (Mdn (IQR), 24.89 (0.09) at 1 h to 24.83 (0.05) at 172 h). Table 3 presents the one-way ANOVA rank test results for comparative differences between the regular and experimental BRMs at four different time intervals. The differences were calculated for mean rank scores, which showed that the lowest ranks based on medians were allotted to Flexitime Bite at all time intervals except at 72 h (MRS 27.17) and 168 h (MRS 28.80), where the regular BRM had the lowest ranks. This indicates that at 72 and 168 h, Flexitime Bite shows more accuracy than the regular material. The post hoc Dunn test results are presented in Table 4, which depicts a clear picture of the significance of differences. Among the three scannable BRMs, at 1 h only Virtual CADBite had accuracy that was more significant than regular BRM. It therefore can be concluded that at 1 h, only Gp SV had significantly more accuracy than regular, while it maintained significant accuracy throughout all time intervals. The only material that showed more significant accuracy than regular at 168 h was Occlufast CAD. When differences within scannable groups were analysed, the Virtual CADBite showed more significant accuracy than Flexitime bite at all time intervals while having no difference with Occlufast CAD, indicating that both Gp SO and Gp SV are equally accurate.

### 3.3. Transparent BRMs (Maxill Bite, Charmflex Bite, Defend ClearBite)

All three transparent BRMs showed fewer linear changes than the regular BRM at all time intervals, with the highest linear changes at 168 h (Gp TC and Gp TD Mdn 24.91 (0.05)) being less than the changes seen in the regular BRM at the 1 h interval (24.91 (0.09)). This indicates that when compared with the regular BRM, they tend to be more accurate at all time intervals (Table 2). Maxill bite showed no change between 72 and 168 h, while Charmflex bite showed no changes between 1 h and 24 h. The differences at various time intervals were considered to be statistically significant when compared with the control (Table 3). The mean rank scores across all time intervals for transparent BRMs ranged from 58.3 (TD1) to 71.77 (Gp TM128), with overall ranks being less than those obtained by the scannable BRMs. The post hoc Dunn test results shown in Table 5 indicate that at 1 h, all transparent BRMs, despite having fewer linear changes than the regular BRM, do not show these differences to be significant, which means that they are equally accurate at the 1 h interval. However, Maxill bite was more significantly accurate at 24 h than the regular, with the other two being equal to the regular. At 72 and 168 h, all three materials show significant accuracy differences from the control, indicating that a BRR to be used at later time should be made of transparent BRM. There were no significant differences between the three transparent materials, which indicates that all three are equally accurate.

### 3.4. Comparison Between Scannable and Transparent

Table 5 presents the comparative results between the various subgroups of scannable and transparent BRMs. Flexitime bite (scannable) was the only material that showed significant linear accuracy differences against Maxill Bite (transparent) at all time intervals and against Charmflex Bite (transparent) at 24, 72, and 168 h. These results indicate that between the scannable and transparent materials, Flexitime Bite had significantly lower accuracy than Maxill Bite and Charmflex bite.

### 3.5. Influence of Time

At 1 h, all materials were equally accurate when compared with the regular material, barring Virtual CADBite, which was significantly more accurate than the regular. At 1 h, Virtual CADBite and Maxill Bite were significantly more accurate than Flexitime Bite. At 24, 72, and 168 h, Virtual CADBite and Maxill Bite were significantly more accurate than the regular, while one scannable (Virtual CADBite) and two transparent (Maxill Bite and Charmflex Bite) were significantly more accurate than Flexitime Bite (scannable). At 72 h, except Occlufast CAD and Flexitime Bite, all other materials were significantly more accurate than regular. At 168 h, all materials except Flexitime bite were significantly more accurate than regular. Charmflex bite (transparent) continued to remain significantly more accurate than Flexitime bite (scannable) at 72 and 168 h.

### 3.6. Dimensional Changes (Millimetres, Percentage) (Table 5)

Table 5 presents a comprehensive tabulation of the changes in dimensions of each subgroup when compared against the regular material and against the original values on the die. When compared against the original values of the die, all material groups and their respective subgroups showed a decrease in the linear dimensions with the least overall decrease ranging at different time intervals. SV1, SV24, TM1, and TD1 were the subgroups that showed the least dimensional changes (0.06 mm) while the subgroup of regular material (R168) showed the highest decrease in linear dimensions (0.17 mm). The clinical threshold of linear changes either in the form of expansion (+) or contraction (−) is 0.11 mm (0.44%) for centric relation changes, which was not fulfilled by Gps R24, R72, R168, SF24, SF72, SF168. Results of dimensional changes when compared against the regular (Occlufast rock) show all subgroups had an increase (expansion) in linear dimensions against their respective control values. The overall interpretation indicates that with each time interval, the differences between the regular BRM and others increase, with the highest differences observed in Gp TM168 (+0.09 mm, +0.36%).

## 4. Discussion

This study aimed to evaluate the linear accuracy of scannable and transparent BRMs against a standard BRM (Occlufast rock) and to estimate the influence of various clinically recommended mounting times (1, 3, and 7 days) on the dimensions. The study also aimed to estimate whether these linear changes are within the clinically acceptable threshold for errors in mounting.

Although we did not specifically test other chemically dissimilar BRMs, our results corroborate those of previous studies that have demonstrated improved dimensional accuracy compared with resin-based [57], zinc oxide eugenol-based [28,38,41], impression plaster-based [27], wax/modified wax [27,30,33,35,36], and polyether-based IORs [28,29,30,31]. The reason for such interpretation is that while most of these studies have also used the similar ADA-specified die, the values for various BRMs when compared with those obtained in our study indicate that the VPS-based scannable and transparent BRMs used in this study have more linear accuracy than non-VPS-based BRMs. Also, when compared with commercial BRMs based on VPS, some researchers have shown that polyether provides more accurate dimensions [37,38,39,40]. On the other hand, research comparing polyether and VPS-based BRMs have also found that the latter maintain better dimensional accuracy over longer durations. In one of these studies, the author used a rectangular die with an increased length [37] and reported 1.5 to 2.4 mm changes in four different conventional VPS-based BRMs compared with 0.80 mm for polyether at 1 h, which further increased after 24 h (PVS 3.4 to 4.0, polyether 2.20) with the differences between VPSs being significant at these times. At 48 and 72 h, however, the differences between the VPS brands were not significant. Another study [38] used an ADA-specified die similar to our study and reported fewer linear changes in polyether at 1 h (0.011%), 24 h (0.012%), and 48 h (0.0127%) when compared with VPS BRMs (1 h (0.012%), 24 h (0.014%), 48 h (0.015%), 72 h (0.016%)). Our study shows that the linear dimensional changes for all scannable VPS BRMs were in the range of 0.24% to 0.56% when compared against the die dimensions. The differences may be attributed to the differences in arithmetic formula used by those authors. Almost identical to the results shown by Lozano F et al. [32] for reduced linear changes at various time intervals are our own results; they found a linear change of 0.12 at 1 h, 1 day, and 7 days, which dropped to 0.072 at the 22nd day. No change in VPS-based IOR was reported by him throughout the first week. According to our study results, all brands of scannable and transparent BRMs did not show any significant changes in accuracy when compared with each other (Table 4), thus indicating that all materials showed consistent accuracy irrespective of types and commercial brands. Out of all the studies that used the ADA-specified die to determine linear changes, only two looked at the effects after seven days or longer [41,42]. The values found for both scannable and transparent at 1 h are lower than those in previous research [22,33,37,41], but they are in the same vicinity as those observed in other studies [30,38,42]. When comparing the dimensional changes with the original die at different time intervals, our values and interpretation (decrease in length) fall in accordance with other, similar research [22,33,37,39,41,42].

The accuracy of VPS BRMs is attributed to multiple factors that include its ability to compensate for dimensional changes within the matrix. The initial matrix expansion is offset by subsequent shrinking. The H-radical creation rate and vinyl-polysiloxane monomer consumption rate may explain why large-molecular-weight, monofunctional monomers have a lower vinyl reaction capability than smaller or branching multifunctional monomers. H-radicals may pair to form H2, which will expand initially [58]. Alternatively, dimethacrylate-based BRMs generate early-phase setting exothermic heat, which speeds up the setting reaction, despite certain groups (CC reactive) of unpolymerized (smaller monomer molecular weight) material being more in dimethacrylates than VPS [59]. Thus, the shrinkage conversion rate affects BRMs independent of the monomer system or reinforcing particles [58]. Since dimensional stability is crucial to the clinical performance of BRMs, and post-curing reactions can take days to reach equilibrium, a material with a low percentage of maximum shrinkage value that is quickly reached and stabilised after setting offers significant clinical benefits. Extended post-setting conversion might compress materials, reducing their dimensional stability. Chun et al. [59] found that dimethacrylate-based interocclusal recording materials shrank the most, followed by polyether and polyvinylsiloxane.

Our results show all materials decreased in the linear dimensions, indicating shrinkage of the BRMs. Elastomers set by continuous crosslinking polymerisation, which densifies molecules and shrinks macroscopically [60]. Carbon double bonds shrink immediately, replacing van der Waals inter-molecular space with intra-molecular covalent bonds [61], changing polymerisation dimensions and density [62]. After 30 min, elastomeric materials polymerise, negatively changing dimension after 24 h. After severe shrinkage, “rebound” or viscoelastic stress release causes elastomers to grow [63]. Due to monomer size variations before polymerisation, polymerisation shrinkage strain should be smaller than monomethacrylate [64]. Polyether and PVS-based BRMs exhibit low setting shrinkage strain, making them ideal bite recording materials. Polymerisation shrinkage is also affected by the viscosity of the elastomer; a low viscosity leads to a large shrinkage strain after polymerisation [57,61]. The viscosity of the substance is contingent upon the length of the siloxane chains, which subsequently determines the molecular weight. In terms of properties, elasticity and compression resistance vary between a VPS-based impression material and a VPS-based BRM. Limiting cross-connecting between polymer chains reduces BRMs’ elasticity, notably under stress (occlusal pressures) [65]. VPS BRM has minimal flexibility, to enable clinicians to remove it from teeth, but does not distort the occlusal record if the patient exerts severe occlusal stresses. PVS BRM has the highest compression resistance (1000 g load) among other materials like polyether, impression paste (ZnOE), and wax [37]. Occlusal forces degrade elastic interocclusal records (PVS and polyether), although PVS recovers 98–100% once the load is removed [66]. Clinically, low flexibility may reduce tooth removal resistance. Limiting elasticity makes bite records harder to verify owing to the higher initial resistance and brittleness [38]. Interocclusal recording materials may also be distorted by compression during articulation. Mounting pressure should be minimal, and deformation depends on material thickness and stiffness. The biting record should be thin and not distorted when mounting castings to the articulator [67,68]. PVS-based bite registration materials provide greater dimensional stability in variable thicknesses [69]. Disinfecting agents and storage media also influence the accuracy of most dental materials, especially the resin-based [70].

Our study results on linear accuracy show that both scannable and transparent BRMs showed significantly more accuracy than the conventional or regular BRM. These findings substantiate the study results of Yazigi C et al. [26], who found significantly more vertical accuracy in scannable BRMs than conventional BRMs. With both vertical and linear accuracy more accurate than conventional, one may conclude that scannable BRMs are accurate despite their compositional alterations to suit scanning. According to our study, the transparent were more accurate than scanning or regular with less deviation from the die dimensions. Therefore, they are a better alternative to regular BRMs for conventional prosthodontic and restorative procedures.

### Strength and Limitations

The majority of investigations compare chemically and physically distinct materials, which puts a confusing influence into the study findings, and very few studies have examined a single IOR material. This research looks at many varieties of VPS, a chemically and physically comparable material with a track record of dimensional precision in IOR applications. Due to its in vitro methodology, the research does not account for many of the variables that could have an impact in a clinical setting. On the other hand, these factors cloud the picture and do not ultimately dictate a material’s specific capabilities. Additional caveats include the following: first, that the findings are specific to the experimental settings and cannot be applied to other situations; second, that there are still other kinds of IOR materials that need to be studied; and third, that vertical precision was not considered in this work. Bias related to methodology include certain instructions not being correctly interpreted and applied.

## 5. Conclusions

Based on the results of this in vitro study, one many conclude that all BRMs, including the regular one, showed linear disparities at all time intervals, which grew throughout time but decreased thereafter. Virtual CADBite, Maxill Bite, and Defend Clearbite were less accurate than the standard and original die among scannable and transparent BRMs. Storage time considerably affects linear accuracy as unfavourable for all studied BRMs, with transparent BRMs having fewer adverse reactions than normal and scannable BRMs. Flexitime bite had linear alterations below the clinical threshold that were within acceptable limits among experimental scannable and transparent BRMs. The study recommends further investigations to determine the influence of disinfectants, contamination, and oral thermal fluctuations on scannable and transparent BRMs, preferably through clinical studies.

## Figures and Tables

**Table 1 polymers-17-00052-t001:** List of materials and instruments used in the study.

Material	Manufacturer	Types
Occlufast Rock	Zhermak, Badia Polesine, Italy	Lot No.: 419353 (Conventional BRM) WT = 30 s, ST = 60 s, H = 95, DR = 20 μm, DS = 7 D
Occlufast CAD	Zhermak, Badia Polesine, Italy.	Lot No.: 404891; CAD BRM (scannable) WT = 30 s, ST = 60 s, H = 95, DR = 20 μm, DS = 7 D
Virtual CADBite	Virtual CADBite, ivoclar vivadent, Amherst, NY, USA.	Lot No.: ZL09W7 CAD BRM (no contrast medium required while scanning) WT = 30 s, ST = 45 s, H = 85, DS = 1.5% linear change, DR = 2 μm.
Flexitime Bite	Kulzer, South Bend, IN, USA.	Lot No.: K010133 (CAD BRM) WT = 30 s, ST = 60 s, H = 90, DS = 1.5% linear change, DR = 2 μm.
Maxill Bite	Maxill, Cortland, OH, USA.	Lot No.: 85500522 (Clear BRM) WT = 30 s, ST = 75 s
Charmflex Bite	Nongshim-ro, Gunpo-si, Gyeonggi-do, Republic of Korea.	Lot No.: 58123003 (Clear BRM) WT = 30 s, ST = 60 s, ER = 99.9%, H = 90 Dispenser D2 (50 mL cartridge)
Defend ClearBite	Defend, Wendt Street Algonquin, IL, USA.	Lot No.: 85500123 [Clear BRM (regular set)] WT = 60 s, ST = 60 s, H = 60–65, Mousse-like and Thixotropic, VPS Visibility for voids or bubbles
MD 520	Durr Dental, Kornwestheim, Germany, Lot No.: 2214103	Combination of aldehydes, quaternary ammonium cations, special surfactants, and excipients in aqueous solution; 100 g MD 520 contain 0.5 g glutaraldehyde, 0.25 g alkyl benzyl dimethyl ammonium chloride. Disinfection time = 10 M; pH: 4.3 ± 0.5

Abbreviations: BRM = bite registration material, WT = working time, ST = setting time, ER = elastic recovery, H (shoreA) = hardness, DR = detail reproduction, DS = dimensional stability, D = days, VPS = vinyl polysiloxane, M = minutes, CAD = computer-aided diagnosis (scannable BRM); Clear BRM = transparent material that shows voids and defects; Compositions: scannable–Vinyl polysiloxane (20–30%), Methylhydrogensiloxane (5–10%), Organoplatinic complex (0.01–0.05%), Silicon dioxide (40–50%), Pigment (10–20%), Food dyes/aroma (0.1–0.5%); Transparent–Polydimethylsiloxane vinyl terminated (20–40%), Dimethylsiloxane copolymer (1–10%), Quartz silica (30–50%), Polysiloxane (<5%).

**Table 2 polymers-17-00052-t002:** Comparative differences in the median values and interquartile ranges of standard and six novel addition vinyl polysiloxane-based (scannable and transparent) bite registration materials observed at four different time periods.

Bite Registration Material (BRM)	BRM Type	Group	Time Interval (Hours)	*n*	Median	IQR	Minimum	Maximum
**Regular** **(Code R)** **(*n* = 60)**	Occlufast Rock	Gp R	R1	15	24.89	0.090	24.81	24.97
			R24	15	24.87	0.079	24.81	24.95
			R72	15	24.86	0.080	24.80	24.95
			R168	15	24.83	0.050	24.79	24.91
**Scannable** **(Code S)** **(*n* = 180)**	Occlufast CAD **(Code O)**	Gp SO	SO1	15	24.93	0.060	24.87	24.97
			SO24	15	24.92	0.060	24.87	24.97
			SO72	15	24.91	0.059	24.86	24.96
			SO168	15	24.90	0.060	24.85	24.95
	Virtual CADBite **(Code V)**	Gp SV	SV1	15	24.94	0.030	24.88	25.00
			SV24	15	24.94	0.030	24.87	24.99
			SV72	15	24.93	0.039	24.87	24.97
			SV168	15	24.91	0.070	24.77	24.97
	Flexitime Bite **(Code F)**	Gp SF	SF1	15	24.89	0.090	24.80	24.94
			SF24	15	24.88	0.079	24.80	24.93
			SF72	15	24.87	0.090	24.80	24.93
			SF168	15	24.86	0.080	24.79	24.92
**Transparent** **(Code T)** **(*n* = 180)**	Maxill Bite **(Code M)**	Gp TM	TM1	15	24.94	0.030	24.89	24.98
			TM24	15	24.93	0.029	24.89	24.97
			TM72	15	24.92	0.030	24.89	24.96
			TM168	15	24.92	0.030	24.88	24.95
	Charmflex Bite **(Code C)**	Gp TC	TC1	15	24.93	0.059	24.84	24.96
			TC24	15	24.93	0.050	24.84	24.96
			TC72	15	24.92	0.050	24.84	24.96
			TC168	15	24.91	0.050	24.83	24.95
	Defend ClearBite **(Code D)**	Gp TD	TD1	15	24.94	0.050	24.89	24.97
			TD24	15	24.93	0.040	24.88	24.96
			TD72	15	24.92	0.050	24.87	24.95
			TD168	15	24.91	0.050	24.86	24.94

Abbreviations: BRM = bite registration material; Gp = group; N = number of specimens; IQR = Interquartile range. Interpretation of Groups: S = scannable (computer-assisted diagnosis) bite registration material; T = transparent bite registration material.

**Table 3 polymers-17-00052-t003:** One-way ANOVA (Kruskal–Wallis rank test) results for median values of linear dimensional accuracy observed in various bite registration materials at four different time intervals.

BRM Types	Time Intervals (Hours) Sub Groups	*n*	MRS	H Statistic	*p* Value
1 h	R1	15	37.4	26.6251	0.00017 *
	SO1	15	53.87		
	SV1	15	71.23		
	SF1	15	25.67		
	TM1	15	69.63		
	TC1	15	54.9		
	TD1	15	58.3		
24 h	R24	15	31.5	33.111	0.0000 *
	SO24	15	57.07		
	SV24	15	73.13		
	SF24	15	25.13		
	TM24	15	70.1		
	TC24	15	60.2		
	TD24	15	58.83		
72 h	R72	15	23.67	37.5521	0.0000 *
	SO72	15	56.03		
	SV72	15	72.17		
	SF72	15	27.17		
	TM72	15	68.83		
	TC72	15	63.13		
	TD72	15	60.00		
168 h	R168	15	20.20	39.1352	0.0000 *
	SO168	15	58.77		
	SV168	15	65.53		
	SF168	15	28.80		
	TM168	15	71.77		
	TC168	15	63.67		
	TD168	15	62.27		

Abbreviations: R = standard (control group); S = scannable CAD BRM; SO = scannable Occlufast CAD; SV = scannable Virtual CADBite; SF = scannable Flexitime CAD Bite; TM = transparent Maxill Bite; TC = transparent Charmflex Bite; TD = transparent Defend ClearBite; p = probability value; MRS = mean rank score; H = difference between two or more groups of an independent variable on a continuous dependent variable. Interpretation of Groups: S = scannable CAD bite registration material; T = transparent bite registration material. Time intervals: 1 h, 24 h, 72 h, 168 h [38,39,41]. Statistical Interpretation: test employed, one-way ANOVA on ranks (Kruskal–Wallis H test); level of the degree of significance was determined based on the value of *p* < 0.05; * = significant.

**Table 4 polymers-17-00052-t004:** Post hoc test results for multiple group comparison showing mean rank differences (MRDs) and their levels of significance when compared with the control and between various subgroups.

Subgroup	Compared Against	1 h		24 h (1 day)		72 h (3 days)		168 h (7 days)	
		MRD	*p*-Value	MRD	*p*-Value	MRD	*p*-Value	MRD	*p*-Value
R1	SO1	−16.4667	0.137	−25.5667	0.0201	−32.3667	0.0034	−38.5667 *	0.0005 *
	SV1	−33.8333 *	0.0022 *	−41.6333 *	0.0001 *	−48.5 *	0.0000	−45.3333 *	0.0000 *
	SF1	11.7333	0.2893	6.3667	0.5627	−3.5	0.7522	−8.6	0.4378
	TM1	−32.2333	0.0036	−38.6 *	0.0004 *	−45.1667 *	0.0000 *	−51.5667 *	0.0000 *
	TC1	−17.5	0.114	−28.7	0.0090	−39.4667 *	0.0003 *	−43.4667 *	0.0000 *
	TD1	−20.9	0.0590	−27.3333	0.0129	−36.3333 *	0.0010 *	−42.0667 *	0.0001 *
SO1	SV1	−17.3667	0.1168	−16.0667	0.1506	−16.1333	0.1455	−6.7667	0.5415
	SF1	28.2	0.0108	31.9333	0.0042	28.8667	0.0092	29.9667	0.0068
	TM1	−15.7667	0.1545	−13.0333	0.2436	−12.8	0.2482	−13	0.2408
	TC1	−1.0333	0.9256	−3.1333	0.7792	−7.1	0.5218	−4.9	0.6584
	TD1	−4.4333	0.6889	−1.7667	0.8744	−3.9667	0.7204	−3.5	0.7522
SV1	SF1	45.5667 *	0.0000 *	48 *	0.0000 *	45 *	0.0000 *	36.7333 *	0.0009 *
	TM1	1.6	0.8851	3.0333	0.7861	3.3333	0.7636	−6.2333	0.5738
	TC1	16.3333	0.1402	12.9333	0.2472	9.0333	0.4151	1.8667	0.8663
	TD1	12.9333	0.2428	14.3	0.2007	12.1667	0.2723	3.2667	0.7682
SF1	TM1	−43.9667 *	0.0000 *	−44.9667 *	0.0000 *	−41.6667 *	0.0001 *	−42.9667 *	0.0001 *
	TC1	−29.2333	0.0082	−35.0667 *	0.0017 *	−35.9667 *	0.0011 *	−34.8667 *	0.0016 *
	TD1	−32.6333	0.0032	−33.7	0.0025	−32.8333	0.0030	−33.4667	0.0025
TM1	TC1	14.7333	0.1833	9.9	0.3757	5.7	0.6071	8.1	0.4649
	TD1	11.3333	0.3060	11.2667	0.3134	8.8333	0.4255	9.5	0.3914
TC1	TD1	−3.4	0.7588	1.3667	0.9027	3.1333	0.7774	1.4	0.8995

Abbreviations: R = standard (control group); S = scannable CAD BRM; SO = scannable Occlufast CAD; SV = scannable Virtual CADBite; SF = scannable Flexitime CAD Bite; TM = transparent Maxill Bite; TC = transparent Charmflex Bite; TD = transparent Defend ClearBite; p = probability value; MRD = mean rank difference; Interpretation of Groups: S = scannable CAD bite registration material; T = transparent bite registration material. Statistical Interpretation: test employed, one-way ANOVA on ranks (Kruskal–Wallis H test); post hoc test, multiple comparison (Dunn test) after Bonferroni’s correction (*p* value/*n*). All significant values denoted as * are significant at the *p* value of ≤ 0.0023. (Corrected α = α/m = 0.05/21 = 0.002381, where α is the *p* value and m is the number of total subgroups).

**Table 5 polymers-17-00052-t005:** Dimensional variations (millimetres, percentage) between the original die measurements and studied bite registration materials (regular, scannable, transparent).

Materials	IOR Types	Subgroup Codes	N	Median	Dimensional Change					
					Against Die			Against Regular BRM (1 h)		
	(Y)				mm	Percent	Status	mm	Percent	Status
Die		OD (X)	3	25.00	25.00	0%	↔	0.0	0.0	↔
Regular BRM (Code R) (n = 60)	Occlufast Rock	R1(X,Y)	15	24.89	−0.11	−0.44	↓	0.0	0.0	↔
		R24 (Y)	15	24.87	−0.13	−0.52	↓	0.0	0.0	↔
		R72 (Y)	15	24.86	−0.14	−0.56	↓	0.0	0.0	↔
		R168 (Y)	15	24.83	−0.17	−0.68	↓	0.0	0.0	↔
Scannable BRM (Code S) (n = 180)	Occlufast CAD (Gp CO)	SO1(X,Y)	15	24.93	−0.07	−0.28	↓	+0.04	+0.16	↑
		SO24	15	24.92	−0.08	−0.32	↓	+0.05	+0.20	↑
		SO72	15	24.91	−0.09	−0.36	↓	+0.05	+0.20	↑
		SO168	15	24.90	−0.10	−0.4	↓	+0.07	+0.28	↑
	Virtual CADBite (Gp CV)	SV1(X,Y)	15	24.94	−0.06	−0.24	↓	+0.05	+0.20	↑
		SV24	15	24.94	−0.06	−0.24	↓	+0.07	+0.28	↑
		SV72	15	24.93	−0.07	−0.28	↓	+0.07	+0.28	↑
		SV168	15	24.91	−0.09	−0.36	↓	+0.08	+0.32	↑
	Flexitime Bite (Gp CF)	SF1(X,Y)	15	24.89	−0.11	−0.44	↓	0.0	0.0	↔
		SF24	15	24.88	−0.12	−0.48	↓	+0.01	+0.04	↑
		SF72	15	24.87	−0.13	−0.52	↓	+0.01	+0.04	↑
		SF168	15	24.86	−0.14	−0.56	↓	+0.03	+0.12	↑
Transparent BRM (Code T) (n = 180)	Maxill Bite (Gp TM)	TM1(X,Y)	15	24.94	−0.06	−0.24	↓	+0.05	+0.20	↑
		TM24	15	24.93	−0.07	−0.28	↓	+0.06	+0.24	↑
		TM72	15	24.92	−0.08	−0.32	↓	+0.06	+0.24	↑
		TM168	15	24.92	−0.08	−0.32	↓	+0.09	+0.36	↑
	Charmflex Bite (Gp TC)	TC1(X,Y)	15	24.93	−0.07	−0.28	↓	+0.04	+0.16	↑
		TC24	15	24.93	−0.07	−0.28	↓	+0.06	+0.24	↑
		TC72	15	24.92	−0.08	−0.32	↓	+0.06	+0.24	↑
		TC168	15	24.91	−0.09	−0.36	↓	+0.08	+0.32	↑
	Defend ClearBite (Gp TD)	TD1(X,Y)	15	24.94	−0.06	−0.24	↓	+0.05	+0.20	↑
		TD24	15	24.93	−0.07	−0.28	↓	+0.06	+0.24	↑
		TD72	15	24.92	−0.08	−0.32	↓	+0.06	+0.24	↑
		TD168	15	24.91	−0.09	−0.36	↓	+0.08	+0.32	↑

Abbreviations: R = standard (control group); S = scannable CAD BRM; SO = scannable Occlufast CAD; SV = scannable Virtual CADBite; SF = scannable Flexitime CAD Bite; TM = transparent Maxill Bite; TC = transparent Charmflex Bite; TD = transparent Defend ClearBite; p = probability value; MRS = mean rank score; H = difference between two or more groups of an independent variable on a continuous dependent variable. Interpretation of Groups: S = scannable CAD bite registration material; T = transparent bite registration material; dimensional change D (%) = (X − Y)/X × 100, where X is the original standard measurement in the die and Y is the observed average measurements of the samples in a particular group; dimensional change D (mms) = X − Y. Symbols (↔ = Same; ↑ = increase; ↓ = decrease).

## Data Availability

All relevant data have been presented within the article; however, the raw data files are available from the corresponding author and can be available upon reasonable request.

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
