# Peer review of "Comparative Assessment of the Influence of Various Time Intervals upon the Linear Accuracy of Regular, Scannable, and Transparent Vinyl Polysiloxane-Based Bite Registration Materials for Indirect Dental Restoration Fabrication"

_polymers, 2024, doi:10.3390/polym17010052_

Round 1

Reviewer 1 Report

Comments and Suggestions for Authors

Dear Authors,

Manuscript title: “Comparative Assessment of the Influence of Various Time Intervals upon Linear Accuracy of Regular, Scannable, and Transparent Vinyl Polysiloxane Based Bite Registration Materials for Indirect Dental Restoration Fabrication”. Your study focuses on an important topic, and I appreciate the effort you have put into your work.

After reviewing your manuscript, I found the study technically sound; however, I believe some aspects could be further refined to strengthen its impact. Research on VPS materials has been widely explored, and the novelty of assessing their linear accuracy over time intervals may be limited, particularly in digital workflows where scanning is expected to occur immediately after capturing the occlusal record. Highlighting how your findings address practical challenges in modern dentistry or comparing VPS materials directly with intraoral scanning techniques could enhance the relevance of your work.

I also noticed that the study mentioned ethics approval, despite the absence of human subjects. Incorporating human participants in a clinical context might add a valuable dimension to your research, potentially making the findings more applicable and interesting for practitioners.

I hope these comments help refine your work. 

Author Response

Reviewer 1

Dear Authors,

Manuscript title: “Comparative Assessment of the Influence of Various Time Intervals upon Linear Accuracy of Regular, Scannable, and Transparent Vinyl Polysiloxane Based Bite Registration Materials for Indirect Dental Restoration Fabrication”. Your study focuses on an important topic, and I appreciate the effort you have put into your work.

Thank you very much for kind appreciation.

After reviewing your manuscript, I found the study technically sound; however, I believe some aspects could be further refined to strengthen its impact.

Definitely we would like to refine the manuscript.

Research on VPS materials has been widely explored, and the novelty of assessing their linear accuracy over time intervals may be limited, particularly in digital workflows where scanning is expected to occur immediately after capturing the occlusal record.

In the introduction section the article mentions in detail different types of digital jaw relations and different forms of interocclusal records involved in each jaw relation. Polyvinyl siloxane based bite registration materials have reported accuracy up to 22 days therefore both manufacturers and studies do not mention or recommend to scan immediately if occlusal records are made up of polyvinyl siloxane.

Highlighting how your findings address practical challenges in modern dentistry or comparing VPS materials directly with intraoral scanning techniques could enhance the relevance of your work.

Our study focusses on physical interoccusal records while intra oral scanning is completely different which may not fall within the scope of this study because the techniques are different. However, we have already mentioned both types and also mentioned “ Currently, however, the most common digital technique utilizes scanning of a polyvi-nylsiloxane IOR in intercuspal position, which provides a two dimensional image that is analyzed with image computer software”

I also noticed that the study mentioned ethics approval, despite the absence of human subjects. Incorporating human participants in a clinical context might add a valuable dimension to your research, potentially making the findings more applicable and interesting for practitioners.

It is a matter of policy of the concerned institute and the university that all research projects including any laboratory in vitro studies need ethical approval before starting the study. We have followed the protocol and the ethical approval in original has been submitted to the journal.

Regarding incorporation of human participants in the study will require a new study design and a new ethical approval. We have mentioned the limitation of in-vitro study in the concluding section. we have also mentioned the drawbacks of conducting such study clinically as it influences the accuracy of concerned material. Our research is purely investigating the ability of the material without subjecting it to influences that could alter its accuracy. Therefore, we request that the study be considered as it has been designed.

I hope these comments help refine your work.

Thank you for refining our work.

Reviewer 2 Report

Comments and Suggestions for Authors

The Authors must see my remarks

Author Response

Reviewer 2

The comments of reviewer 2 have been addressed in the file as suggested by the editor

All changes made within the manuscript have been tracked and are clearly visible.

Reviewer 3 Report

Comments and Suggestions for Authors

This paper presents comparative assessment of the influence of various time intervals upon linear accuracy of different vinyl polysiloxane based bite registration materials for indirect dental restoration fabrication.

I recommend accept this paper in presented form.

Round 2

Reviewer 1 Report

Comments and Suggestions for Authors

My opinion is that this manuscript is not sufficiently relevant to be published.